# Crosslinked Nanocomposite Sodium Alginate-Based Membranes with Titanium Dioxide for the Dehydration of Isopropanol by Pervaporation

**DOI:** 10.3390/molecules25061298

**Published:** 2020-03-12

**Authors:** H.G. Premakshi, Mahadevappa Y. Kariduraganavar, Geoffrey R. Mitchell

**Affiliations:** 1P. G. Department of Studies in Chemistry, Karnatak University, Dharwad 580 003, India; premakshihg161@gmail.com; 2Centre for Rapid & Sustainable Product Development, Polytechnic of Leiria, 2430-028 Marinha Grande, Portugal

**Keywords:** isopropanol, pervaporation, poly(styrene sulfonic acid-co-maleic acid), sodium alginate, titanium dioxide

## Abstract

Sodium alginate (NaAlg) based membranes were prepared using a solution technique, crosslinked with poly(styrene sulfonic acid-co-maleic acid) (PSSA-co-MA). Subsequently, the membranes were modified by the incorporation of 0, 10, 20, 30 and 40% *w*/*w* of titanium dioxide with respect to sodium alginate. The membranes thus obtained were designated as M, M-1, M-2, M-3 and M-4, respectively. An equilibrium swelling experiment was performed using different compositions of the water and isopropanol mixtures. Subsequently, we used a pervaporation cell fitted with each membrane in order to evaluate the extent of the pervaporation dehydration of isopropanol. Among the membranes studied, the membranes containing 40 mass% of titanium dioxide exhibited the highest separation factor(α) of 24,092, with a flux(J) of 18.61 × 10^−2^ kg/m^2^∙h at 30 °C for 10 mass% *w*/*w* of water in the feed. The total flux and the flux of water were found to overlap with each other, indicating that these membranes can be effectively used to break the azeotropic point of water–isopropanol mixtures. The results clearly indicate that these nanocomposite membranes exhibit an excellent performance in the dehydration of isopropanol. The activation energy values obtained for the water permeation were significantly lower than those of the isopropanol permeation, underlining that these membranes have a high separation ability for the water–isopropanol system. The estimated activation energies for total permeation (E_P_) and total diffusion (E_D_) values ranged between 10.60 kJ∙mol^−1^ and 3.96 kJ∙mol^−1^, and 10.76 kJ∙mol^−1^ and 4.29 kJ∙mol^−1^, respectively. The negative change in the enthalpy values for all the membranes indicates that sorption was mainly dominated by Langmuir’s mode of sorption.

## 1. Introduction

Pervaporation has emerged as an alternative to classical distillation processes for the dehydration of alcohols. However, the adoption of pervaporation is limited by the economics of the technology [1]. We can quantitatively evaluate the performance of a membrane in terms of the permeation flux and the level of selectivity. Other aspects which are important include both mechanical strength and chemical stability [2,3,4,5]. Mass transfer in the pervaporation process is based on the solution diffusion mechanism: selective sorption of the feed species on the membrane surface, selective diffusion through the membrane, and desorption into a vapor phase on the permeate side. Considering these steps, it is often found that selective sorption and diffusion in the membrane are the rate-governing steps [6]. These are high when there are strong interactions between the feed mixture and the polymer matrix, since the transport of the permeant is not only dependent on its own driving force but also on the friction of this permeant with other components in the feed and polymer matrix [7].

The separation of mixtures of alcohols has received much attention due to the potential applications in industry. Generally, low-molecular-weight alcohols are miscible with water and form azeotropic mixtures, which cannot be separated easily by traditional distillation [8]. Amongst the numerous alcohols, isopropanol is an industrially important liquid, and forms an azeotrope with water at a composition of isopropanol 87.7 wt% [9]. The pervaporation process can effectively separate azeotropic mixtures while reducing the cost and energy consumption. The technical feasibility of this process largely depends on the membrane and its properties. Most of the studies reported in the scientific literature on pervaporation separation are focused on the use of hydrophilic materials which preferentially absorb the water molecules, leading to high selectivity and permeation flux [10,11,12,13,14,15]. Sodium alginate as a hydrophilic polymer is widely explored in pervaporation as a membrane material for the separation of water from organic mixtures due to its excellent film-forming ability, separation selectivity towards water, and good resistance to organic solvents [16,17,18]. Furthermore, sodium alginate is a natural polysaccharide which can be obtained from sustainable sources. Unfortunately, sodium alginate membranes exhibit substantial swelling in aqueous solutions, which results in a significant decline with time in membrane selectivity as well as mechanical strength due to the presence of large numbers of carboxyl and hydroxyl groups.

Yeom and Lee [19] improved the membrane strength and stability of sodium alginate by crosslinking the membrane with glutaraldehyde. Huang et al. [20] also prepared stabilized sodium alginate membranes using the relatively simple method of ionic crosslinking. Generally, crosslinking limits the mobility of the polymer chains, and this will limit membrane swelling due to the change in the balance between the swelling and the deformation of the network. Such crosslinked membranes show a clear increase in their selectivity, but the crosslinking does not lead to an enhancement of the flux. To overcome this, Kariduraganavar et al. crosslinked the sodium alginate membrane with poly(styrene sulfonic acid-co-maleic acid) (PSSA-co-MA), which limits the degree of swelling while simultaneously increasing both flux and selectivity [21].

Very recently, researchers have developed nanocomposite membranes in order to increase the membrane performance by incorporating nanofillers into a polymeric matrix [22]. Inorganic nanofillers, particularly TiO_2_, demonstrated an excellent performance in membrane fouling mitigation and enhancement of flux owing to their superhydrophilic characters upon irradiation with UV light [23,24,25,26], During the process of UV irradiation, a complex and not fully understood process leads to the reduction of the Ti(IV) state to the Ti(III) state by the photo activated electrons, and the holes oxidize the O^2−^ ions. This leads to a new surface containing absorbed -OH groups, rendering the surface hydrophilic [27]. Such materials have the unique property of attracting water molecules rather than repelling, which is commonly known as superhydrophilicity [28]. Further, it is reported that titanium dioxide can also be used in aggressive conditions because of its high chemical and thermal stability [29].

Based on this background information, we have set out to develop membranes for pervaporation by incorporating different levels of TiO_2_ into a sodium alginate matrix, which was further crosslinked with PSSA-co-MA. The physico-chemical properties of the resulting membranes were studied using Fourier transform infra-red spectroscopy, wide-angle X-ray diffraction, differential scanning calorimetry, thermal gravimetric analysis, scanning electron microscopy, and mechanical testing. The membranes were successfully employed for the separation of water-isopropanol mixtures at different feed compositions. The values of permeation flux and separation selectivity were determined and compared with reported results. The diffusion coefficient and Arrhenius activation parameters were calculated. The results were discussed in terms of the pervaporation separation efficiency of the membranes.

## 2. Results and Discussion

### 2.1. Membrane Characterization

#### 2.1.1. Fourier Transform Infra-red Spectroscopy Studies

Fourier Transform Infra-red (FTIR) spectra of crosslinked membranes are presented in Figure 1. In membrane M, a characteristic strong and broad band occurred at around 3442 cm^−1^, which is attributed to O-H stretching vibrations of the hydroxyl groups. The bands at 1610 cm^−1^ and 1410 cm^−1^ are assigned to asymmetric and symmetric C=O stretching vibrations of maleic acid. Similarly, the multiple bands that appear between 1000 cm^−1^ and 1200 cm^−1^ are assigned to C-O stretching. These bands overlap with the symmetric and asymmetric stretching bands of the S=O groups, which are the characteristic groups of chemically bonded -SO_3_ groups of polystyrene sulfonic acid-co-maleic acid [21]. In addition, a band at 1240 cm^−1^ is observed in the crosslinked membrane, indicating the formation of ether linkages between the -OH of sodium alginate and the -COOH of maleic acid. The intensity of all these bands decreased as the content of titanium dioxide was increased in the membranes, suggesting that the interactions between titanium dioxide and the sodium alginate matrix were enhanced. Upon increasing the titanium dioxide content in the membrane matrix from M-1 to M-4, a band observed in the region 450–800 cm^−1^ became broader. This was mainly attributed to the stretching mode of the Ti-O bond and the enhancement of hydrogen bonding interactions between sodium alginate and the titanium dioxide [24,30]. This indicates that there is a good compatibility between the crosslinked sodium alginate and the titanium dioxide. Thus, the resulting membranes would be better candidates for the enhancement of pervaporation performance.

#### 2.1.2. WAXD Studies

Figure 2 shows the X-ray diffraction patterns of the resulting membranes employed for the PV study; each of the membranes has the same thickness. The data for the crosslinked sodium alginate membrane (M) show a broad peak at 2θ = 13.5° typical of the semi-crystalline sodium alginate, whereas the other membranes (M-1, M-2, M-3, and M-4) display sharp peaks which correspond to the anatase crystalline form of titanium dioxide. The broad feature shown at 2θ~13.5° is not apparent in any of the patterns for the membranes containing titanium dioxide. This suggests that the crystallinity has been drastically reduced by the addition of the titanium dioxide. However, the differential scanning calorimetry (DSC )data (Table 1) show that the crystallinity has been reduced by ~27% in the case of M-4. We can understand these differences if we take account of the dominating effect of the titanium dioxide on the scattering pattern due to its high atomic number. The decrease of crystallinity and the accompanying increase in both the amorphous regions and the accompanying chain flexibility make it easier for small molecules such as water to diffuse through the membrane. This arises in particular as the titanium dioxide particles appear to be uniformly distributed in the crosslinked sodium alginate matrix, which will increase the interaction between the crosslinked sodium alginate and the titanium dioxide. This favors the selective transport of water molecules, which in turn increases the overall pervaporation performance [31,32,33]. The titanium dioxide is present in the membrane as crystalline particles, and these show characteristic peaks at 2θ ≈ 26°, 38°, and 48°. We note that the intensities of the peak at 2 θ = 26° do not follow a simple linear scaling with composition. We attribute some of this to different peak widths, which indicate that the crystalline particles in each sample are not the same size, but it may also reveal that some of the titanium dioxide is present in a different form, reflecting the increased interactions between the sodium alginate and the titanium dioxide. The former is in good agreement with the SEM results, in which the agglomeration of the titanium dioxide particles in the membrane matrix can be clearly seen. The decrease of the crystalline alginate phase and the shrinkage of cells together contribute to making the membrane structure more compact, which contributes to the level of selective permeation of penetrants through the membrane [8].

#### 2.1.3. Thermal Studies

The thermogravimetric curves of the crosslinked membranes are presented in Figure 3. From the thermograms, it is clear that under nitrogen flow, non-oxidative degradation occurred in three stages for all the membranes. The first weight loss occurred between ambient temperature and 110 °C, corresponding to the loss of the physically absorbed water molecules. Most of these absorbed water molecules exist in a bound state rather than in a free molecular state and the water molecules are bound directly to the polymer chain through hydrogen bonds. The weight loss is about 28% for the crosslinked sodium alginate membrane, whereas the nanocomposite membranes exhibited a lower weight loss ranging from 15% to 18%. This indicates that membranes containing titanium dioxide have a lower water-retention capacity. The second stage of decomposition started in a narrow range of temperature from 220 °C to 285 °C. This was attributed to a major weight loss (~18%) due to the decomposition of the sodium alginate matrix, whereas the nanocomposite membranes exhibited a slightly lower loss of around 15%. The third stage of decomposition occurred from 430 °C to 550 °C and corresponded to a weight loss of about 10% to 12%. This was attributed to the decomposition of polystyrene sulfonic acid-co-maleic acid. If we consider the temperature at 50% weight loss as a measuring point, the nanocomposite membranes exhibited temperatures around 10 °C to 20 °C higher than those of the crosslinked sodium alginate membrane. The weight loss was lower with an increasing level of titanium dioxide. In Table 1 we show the glass transition temperature (Tg) and crystallinity values obtained for each membrane using the DSC data. As the proportion of titanium dioxide increased in each membrane the value of Tg fell, as did the fraction of crystallinity.

#### 2.1.4. SEM Studies

As far as can be judged from SEM micrographs, the surfaces of the membrane M were smooth. No voids can be seen for the membranes with lower titanium dioxide content. Scanning electron micrographs of surface and cross-sectional views of the membranes are shown in Figure 4. The nonporous structure of the dense membrane is observed from the cross-section of the membrane. The thickness of membrane was found to be about 40 µm ± 2. It is clearly observed from SEM that the TiO_2_ particles formed on the surface due to the TiO_2_ conglomeration. The distribution of TiO_2_ increased from membrane M-1 to M-4 with increasing content of TiO_2_. These findings indicate that the addition of TiO_2_ nanoparticles has a large effect on the membrane structure, which might provide extra free volumes to the polymer chains, and consequently offer spaces for water molecules to permeate through the membranes. This is in broad agreement with the WAXD results.

### 2.2. Mechanical Properties

The tensile strength and the extensibility of a membrane often indicate its suitability for pervaporation applications. The effect of titanium dioxide on the tensile strength and the elongation at failure of the crosslinked membranes were measured and the results are summarized in Table 1. It could be seen that tensile strength of the membranes increased from membrane M to M-1. This is expected owing to the reinforcing effect of the uniform distribution of titanium dioxide in the crosslinked sodium alginate membrane matrix. However, when the content of titanium dioxide was increased beyond 10%, the tensile strength decreased slightly. This is due to the formation of larger agglomerates of titanium dioxide in the membrane matrix, which were also observed in the SEM and WAXD results. On the other hand, the elongation at failure decreased continuously from membranes M to M-4. We attribute this to the interaction of titanium dioxide with the sodium alginate.

### 2.3. Effects of Feed Composition and Titanium Dioxide on Membrane Swelling

In the pervaporation process, the membrane sorption plays a key role in the separation performance of the membrane, which is generally assessed by studying the level of membrane swelling in different compositions of the feed mixtures. Figure 5 shows the degree of membrane swelling plotted as a function of different compositions of the water-isopropanol mixtures at 30 °C. The degree of swelling increases more or less linearly for all the membranes, with an increase in the fraction of water in the feed. This is due to a greater interaction between water molecules and the membrane.

We attribute this to the greater polarity of water compared with isopropanol, which preferentially interacts with hydrophilic groups of the membrane. The degree of swelling also increased with a higher level of titanium dioxide in the crosslinked sodium alginate membrane. Increasing the amount of filler particles reduces the number of network chains per unit volume, thus reducing the entropic restricting force of the chains against swelling. However, we have found some evidence that the sodium alginate matrix appears to exhibit enhanced interactions with the titanium dioxide particles, and this may serve to increase the effective level of the crosslinking which would act to reduce the degree of swelling. We also note that titanium dioxide induces superhydrophilicity in the membrane matrix by irradiation of UV-light, irrespective of its crystalline nature [34,35].

The mechanism for the photo-induced hydrophilicities of titanium dioxide is due to structural changes. In this mechanism, the Ti(IV) states are reduced to the Ti(III) states via photo-generated electrons and as a result oxygen vacancies are generated through the oxidation of bridging O^2−^ species to oxygen, which leads to a surface containing absorbed hydrophilic -OH groups. Consequently, the adsorption of water molecules is increased, and this in turn becomes responsible for the enhanced degree of swelling with increasing titanium dioxide content in the membrane. The formation of hydrogen bonds became more predominant in the membrane matrix as the content of titanium dioxide was increased, and this was observed in the FTIR data. This weakened the interactions which existed in the crosslinked sodium alginate matrix, and thereby the crystallinity of the crosslinked membrane matrix was decreased (Table 1) while the free volume in the matrix increased. Evidently, the increased free volume and change in membrane structure were responsible for increasing the degree of swelling with increased titanium dioxide content in the crosslinked sodium alginate membrane matrix.

### 2.4. Effects of Feed Composition and Titanium Dioxide on Pervaporation

Figure 6a shows the effects of feed composition and the titanium dioxide content on the total permeation flux for all the membranes at 30 °C. It was found that the total permeation flux was increased significantly for all membranes with increasing water composition in the feed. This is mainly because of the increased interactions between the water molecules and the membranes, since membranes have large number of hydrophilic groups such as -OH, -COONa, -COOH, and -SO_3_H.

Similarly, the permeation flux was increased from membrane M to M-4 at all water compositions through the increase of the titanium dioxide content. This is due to an increased selective interaction between water molecules and the membranes, since the crosslinked sodium alginate membrane contains active groups which are capable of forming electrostatic force and hydrogen bonds with the water molecules. However, this interaction was more prominent for the titanium dioxide-incorporated composite membranes (M-1 to M-4). We attribute this to the superhydrophilicity in the membrane matrix, which enhances the greater interaction between the water molecules and the membranes. As we identified earlier, the decrease of crystallinity of the sodium alginate is another factor for the enhancement of flux. Since the crystalline regions are inaccessible to the permeants, the decrease in the crystallinity of the membranes by the incorporation of titanium dioxide could be responsible for the increase of flux.

In the pervaporation process, the overall selectivity of a membrane is generally explained in terms of the interaction between the membrane and the permeating molecules, the molecular size of the permeating species, and the pore diameter of the membrane [36]. Figure 6b displays the effects of water composition and titanium dioxide content on the selectivity of all the membranes. It was observed that the selectivity of all the membranes decreased drastically with increasing water content in the feed. This is because at higher concentrations of water in the feed, the membranes swell greatly owing to the formation of strong interactions between the membrane and the water molecules.

In contrast, the selectivity of these membranes increased from membrane M to M-4 with the increase in titanium dioxide content in the membrane matrix. Firstly, this is attributed to an increased selective interaction between the membrane and the water molecules owing to the establishment of superhydrophilicity in the membrane matrix. Secondly, there is a decrease in the semicrystalline regions due to crosslinks (Ti-O-C) between the active groups of sodium alginate (-CH_2_OH and -OH) and titanium dioxide (Ti-OH). Lastly, there is covalent crosslinking between the -COOH group of polystyrene sulfonic acid-co-maleic acid, the -OH group of sodium alginate, and the hydrophilic donor groups (-SO_3_H), which contributed to the shrinkage in cell size and enhancement of hydrophilicity. Since the sizes of water molecules are small as compared to isopropanol, water molecules easily and preferentially pass through the membrane [20].

Figure 7a demonstrates the variation of flux and selectivity as a function of titanium dioxide content in the membrane with 10% water in the feed. From this plot, it could be seen that both permeation flux and selectivity increased with increasing titanium dioxide content in the membrane. This is in contrast to the more common trade-off phenomenon existing between flux and selectivity in the pervaporation process and underlines the specific properties of the membranes developed in this work. There is a significant enhancement of hydrophilicity (i.e., superhydrophilicity) caused by the reduction of Ti(IV) to Ti(III) and hydrogen bonding through the incorporation of titanium dioxide in the membrane matrix.

The performance of the membranes in the pervaporation process is generally evaluated based on the permeation of the individual components. The extent of permeation of the individual components was therefore determined by plotting the total flux and the fluxes of water and isopropanol as a function of titanium dioxide content in the membranes with 10% water in the feed (Figure 7b). From the plot, it is clear that the total flux and flux of water overlap with each other, particularly for the titanium dioxide nanocomposite membranes, and that the flux of isopropanol is negligibly small for all the membranes. This confirms that the membranes developed in the present study by the incorporation of titanium dioxide particles demonstrate excellent performance as compared to a crosslinked sodium alginate membrane (M).

### 2.5. Comparison of Pervaporation Performance of Sodium Alginate-Based Membranes

The performance of the sodium alginate-based membranes assessed from the determination of flux, separation selectivity, and pervaporation separation index (PSI) in the separation of water–isopropanol mixtures reported in the literature is summarized in Table 2.

Among the membranes presented in Table 2, the membranes which contained SBA-15, Fe-SBA-15, and Na^+^-MMT exhibited infinite separation factors, but unfortunately the observed fluxes were in the range of 0.11 × 10^−2^ to 1.07 × 10^−2^ kg∙m^2^∙h^−1^, which are very low values in comparison with those recorded for the composite membranes reported in the present work. Similarly, membranes containing MWCNT and NaY zeolite demonstrated excellent fluxes, but the separation selectivities were very low compared with the membrane reported with 40% titanium dioxide. In particular, the sodium alginate membrane containing 3% of polystyrene sulfonic acid-co-maleic acid reported by our group exhibited an excellent selectivity of 22,312, with a flux of 10.53 × 10^−2^ kg∙m^−2^∙h^−1^. However, the membrane with the modification of 40% titanium dioxide detailed in the present work exhibited a separation factor and a flux of 24,092 and 18.61 × 10^−2^ kg∙m^−2^∙h^−1^, respectively. This is very well reflected in the pervaporation separation index, which increased from 2415 to 2583, an increase of ~5%. These observations directly demonstrate that crosslinked sodium alginate membranes modified with titanium dioxide showed improved overall performance in the separation of water–isopropanol mixtures. This is all due to the unique nature of titanium dioxide dispersed in the crosslinked sodium alginate matrix of the membrane. It is interesting to note that the titanium dioxide particles probably make a number of contributions to the performance of the membrane. Although increasing the fraction of titanium dioxide to 40% leads to some aggregation effects which impact the mechanical properties, this does not appear to impact on the influence on the hydrophilicity of the membrane, as Figure 7 and Figure 8 show there is a smooth increase in the flux and selectivity with increasing titanium dioxide.

### 2.6. Diffusion Coefficients

The solution diffusion mechanism plays an important role in understanding the pervaporation process. According to this mechanism, the pervaporation process involves three consecutive steps: sorption of the permeant from the feed liquid to the membrane, diffusion of the permeant in the membrane, and desorption of the permeant to the vapor phase on the downstream side of the membrane [46]. Thus, the permeation rate and selectivity are governed by the solubility and diffusivity of each component of the feed mixture to be separated. However, during the process, the diffusion step controls the transport of penetrants owing to an establishment of a fast equilibrium distribution between bulk feed and the upstream surface of a membrane [47,48].

To understand the mechanism of transport of the molecules, it is important to estimate the diffusion coefficient, *D_i_*, of each of the penetrating molecules. From Fick’s law of diffusion (Equation (1)), the diffusion flux can be expressed as [49]:(1)$$J_{i}=-D_{i}\frac{dC_{i}}{dx}$$
where *J* is the permeation flux per unit area (kg m^−2^ s), *D* is the diffusion coefficient (m^2^ s^−1^), *C* is the concentration of permeant (kg m^−3^), subscript *i* indicates water or isopropanol, and *x* is the diffusion length (m). For simplicity, it is assumed that the concentration profile along the diffusion length is linear. Therefore, *D_i_* can be calculated with the following Equation (2): [50]
(2)$$D_{i}=\frac{J_{i}\delta }{C_{i}}$$
where *δ* is the membrane thickness. The calculated values of *D_i_* at 30 °C are presented in Table 3. As observed in the pervaporation study, the diffusion coefficient of water increased significantly from membrane M to M-4, while the diffusion coefficient for isopropanol remains very low. As discussed previously, this is due to the large amorphous phase and the establishment of hydrogen bonding in the membrane matrix as a consequence of the titanium dioxide. A significant increase in the hydrophilicity of the membrane through the incorporation of titanium dioxide promotes water adsorption into the membrane along with isopropanol. However, the magnitudes of diffusion coefficients of water are much higher in comparison with those of isopropanol, indicating that the membranes developed in the present study are highly selective towards water molecules even with higher concentrations of water in the feed.

### 2.7. Effect of Temperature on the Pervaporation Performance

Temperature is an important operating parameter in the pervaporation process as it affects both sorption and diffusion. Hence, it can significantly affect the performance of the membranes. We studied the effect of the operating temperature on the pervaporation performance of each membrane for water-isopropanol mixtures with 10% water in the feed and the results are presented in Table 4.

The permeation rate was found to increase from 30 °C to 50 °C for all the membranes, while reducing the separation selectivity. We consider two reasons. Firstly, as the temperature increases, the viscosity of the permeating molecules decreases due to decreased cohesive forces between the permeants. Secondly, an increase of thermal energy intensifies the motion of polymer chain segments, possibly creating a higher level of free-volume in the polymer matrix. However, in the present study, the latter reason is ruled out since the experiments were performed well below the glass transition temperature of the crosslinked sodium alginate [51]. Therefore, the viscosity of permeating molecules played a major role in allowing the associated molecules along with the selective permeants. As a result, the permeation of diffusing molecules and the associated molecules through the membrane became easier, leading to an increase of total permeation flux while suppressing the selectivity. This effect prompted us to estimate the activation energies for permeation and diffusion using the following Arrhenius-type Equation (3) [52].
(3)$$J=J_{o}\exp\left(\frac{-E_{x}}{RT}\right)$$
where *J* represents permeation, or diffusion (*D*). J_o_ is a constant, representing a pre-exponential factor of *J_o_* or *D_o_*. *E_x_* represents activation energy for permeation or diffusion depending upon the transport process under consideration, and *RT* is the usual energy term. With increasing feed temperature, the vapor pressure in the feed chamber increases, but the vapor pressure at the permeate side is not affected. This results to an increase of driving force for the transport permeants across the membrane.

The Arrhenius plots of log *J* and log *D* versus 1/T are shown in Figure 8a, b for the temperature dependence of permeation flux and diffusion, respectively. From the least-squares fits of these linear plots, the activation energies for total permeation (E_P_) and diffusion (E_D_) were calculated. In a similar way, activation energies for the permeation of water (E_PW_) and isopropanol (E_PIPA_) and for the diffusion of water (E_DW_) were calculated and the values are presented in Table 5. From this table, we note that the crosslinked sodium alginate membrane (M) exhibits much higher activation energies for the total permeation and for diffusion compared to those for the nanocomposite membranes (M-1 to M-4).

This suggests that both permeation and diffusion processes require more energy for transport of molecules through the crosslinked sodium alginate membrane. However, titanium dioxide nanocomposite membranes consumed less energy because of their increased hydrophilicity and the decreased crystalline domains. Although the activation energy values for permeation are slightly smaller than the activation energies for diffusion for all the membranes, the difference is not significant, suggesting that both sorption and diffusion contribute equally to the pervaporation process. The same trend is also observed for both the permeation and the diffusion of water. In contrast, a significant difference was noticed between water and IPA, and the difference increased with increasing content of titanium dioxide. This further underlines the observation that membranes with higher amounts of titanium dioxide demonstrated greater separation efficiency towards water. Using these values, we calculated the heat of sorption (∆H_s_) (Equation (4)) as:(4)$$∆H_{s}=E_{p}-E_{D}$$

The results are included in Table 5. The heat of sorption (∆Hs) values gives additional information about the transport of molecules through the polymer matrix. This is a composite parameter involving contributions of both Henry’s and Langmuir’s types of sorption. [53]. Henry’s law states that the heat of sorption will be positive for liquid transport, leading to the dissolution of chemical species into that site within the membrane, giving an endothermic contribution to the sorption process. However, Langmuir’s sorption requires the pre-existence of a site in which sorption occurs only by a hole-filling mechanism, giving an exothermic contribution. From the data, it is observed that all the membranes exhibited negative ∆*Hs* values, suggesting that Langmuir’s mode of sorption is predominant during the transport of molecules across the polymer matrix. This indicates the pre-existence of holes, and in view of this, only the hole-filling mechanism predominates. This is expected due to the incorporation of titanium dioxide in the membrane matrix.

## 3. Materials and Methods

### 3.1. Materials

Sodium alginate (medium viscosity grade) (NaAlg), titanium dioxide (0.25–10 µm) (TiO_2_), and isopropanol were obtained from S. D. Fine Chemicals Ltd., Mumbai, India. Polystyrene sulfonic acid-co-maleic acid as a crosslinking agent and as a donor of the hydrophilic -SO_3_H and -COOH groups was obtained from Sigma-Aldrich Chemie, GmbH, Germany. All of these chemicals were of reagent grade and were used without further purification. Double-distilled water was used throughout the study.

### 3.2. Membrane Preparation

We performed an initial screening study based on previously published work and additional experiments which revealed that 4% *w*/*w* aqueous solutions provided the optimum starting point with regard to the final performance of the membrane. At the same time, we determined that the optimum thickness for the membranes was 35–40 µm. Accordingly, a 4% *w*/*w* aqueous solution of sodium alginate was prepared by adding dry sodium alginate to double-distilled water and continuously stirring for 24 h. This mixture was filtered and to the filtrate, 40% *w*/*w* of PSSA-co-MA was added as a crosslinker. The reaction mixture was stirred to form a homogenous mixture and left overnight at room temperature. The resulting homogeneous mixture was cast onto a clean glass plate with the help of a casting knife and allowed to dry at room temperature in a dust free atmosphere for 48 h. It was possible to peel off the dried membrane which was then annealed at 120 °C for 2 h in order to induce crosslinking. The membrane prepared using this methodology was designated as M.

For the preparation of composite membranes, a specific amount of titanium dioxide was added into the above homogeneous mixture. This was stirred to produce a uniform mixture and then left overnight at room temperature to remove air bubbles. That mixture was then subjected to the same procedure for crosslinking as described above. The proportion of titanium dioxide with respect to sodium alginate was varied at 10%, 20%, 30%, and 40% *w*/*w*, and the quantity of sodium alginate was kept constant. The resulting membranes were designated as M-1, M-2, M-3, and M-4, respectively, reflecting the fractions of titanium dioxide. The 40% *w*/*w* was the highest amount of titanium dioxide that it was possible to incorporate and still produce a coherent membrane using the method described here. These membranes were subsequently subjected to the same thermal treatment around 120 °C for 2 h and used for membrane M. The proportion of the crosslinking agent (PSSA-co-MA) was kept constant for all the membranes and only the fraction of TiO_2_ in the crosslinked NaAlg membrane (M) was varied. This scheme for the preparation of nanocomposite membranes is illustrated in Figure 9. The thickness of these membranes measured at different points using a Peacock dial thickness gauge (Model G, Ozaki Mfg. Co. Ltd., Tokiwadai, Itabashi-Ku, Tokyo Japan) with an accuracy of ± 2 μm was found to be 40 ± 2 μm.

### 3.3. Fourier Transform Infrared Spectroscopy (FTIR)

The fraction of TiO_2_ in each crosslinked sodium alginate membrane was confirmed using infra-red spectroscopy (Nicolet FTIR, Impact-410, Fall River, Massachusetts, USA. Membrane samples were ground and mixed with potassium bromide to prepare pellets using a hydraulic pressure of 400 kgcm^2^, and the spectra were recorded in the range of 400–4000 cm^−1^. For each scan, the mass of the membrane sample and potassium bromide were kept constant in order to directly reveal the changes in the intensities of the characteristic peaks with the fraction of titanium dioxide.

### 3.4. Wide-Angle X-Ray Diffraction (WAXD)

Wide-angle X-ray diffraction data were obtained for each membrane at room temperature using a Philips analytical X-ray system equipped with a copper target X-ray tube operated at 40 kV and 30 mA. A nickel filter was used in the incident beam to yield Cu-Kα radiation. Data were obtained in the reflection mode for a single membrane film over an angular range of 5 to 50° 2θ at a scan speed of 8° min^−1^. The membranes, which were prepared for the pervaporation study, were directly used for the WAXD measurements.

### 3.5. Thermal Analysis

Each membrane was subjected to thermogravimetric analysis (TA Instrument Model DSC Q 20, New Castle, DE, USA under a nitrogen atmosphere over the temperature range of 25–550 °C at a heating rate of 10 °C min^−1^. We evaluated the level of crystallinity and the glass transition for each membrane. The thermal properties of each membrane were investigated using a Perkin-Elmer differential scanning calorimeter (SDT Q600, TA Instruments-Waters LLC, New Castle, DE USA. Samples of ~8–10 mg were heated from ambient temperature to 120 °C at a heating rate of 10 °C/min.

### 3.6. Scanning Electron Microscopy (SEM)

The morphology of each vacuum dried membrane was examined using a scanning electron microscope (JEOL, JSM-400 Å, Tokyo, Japan). All samples were coated with a conductive layer (400 Å) of sputtered gold. The membranes, which were prepared for the PV study, were directly used for recording the SEM images after gold sputtering.

### 3.7. Tensile Properties

The mechanical properties of each membrane were measured at 25 °C using a universal testing machine (Hounsfield, H10KS) with a crosshead speed of 50 mm min^−1^ and a gauge dimension of 25 mm × 50 mm. The results reported are an average of measurements made on three separate specimens.

### 3.8. Swelling Measurement

Vacuum-dried membranes were subjected to equilibrium sorption measurements by soaking in feed stock with different compositions of water–isopropanol mixtures at 30 °C for 24 h to attain equilibrium swelling. The swollen membranes were taken out at regular intervals, wiped with tissue paper carefully to remove the surface adhered solvent, and weighed as quickly as possible on a digital microbalance (Mettler, AE 240, CH-8606, Greifensee, Switzerland) with an accuracy of ± 0.01 mg. The results reported are an average of three tests; based on the weight of the adsorbate, the degree of swelling (*DS*) was calculated using the following Equation (5):(5)$$DS\left(\%\right)=\left(\frac{w_{s}-w_{d}}{W_{d}}\right)\times 100$$
where *W_s_* and *W_d_* represent the weight of the swollen and the dry membranes, respectively.

### 3.9. Pervaporation Experiments

The pervaporation cell was constructed from two cylindrical half cells made of stainless steel fastened together by nuts and bolts as shown in Figure 10. The membrane was placed on a perforated stainless steel plate at the junction of two half cells. The capacity of the feed cell was ~250 cm^3^ and the effective surface area of the membrane in contact with the feed mixture was 34.23 cm^2^. The feed temperature was maintained by circulating water through the outer cell jacket of the cell; the experiments were carried out at 30 °C, 40 °C, and 50 °C. For all the measurements, the downstream pressure was maintained at 10 Torr (1.33 × 10^3^ Pa) using a two-stage vacuum pump (Toshniwal, Chennai, India). The water composition in the feed mixture was varied from 5% to 25% *w*/*w* by considering the azeotropic point of the water-isopropanol mixture. Before performing the PV experiments, the test membrane was allowed to attain equilibrium swelling with a known volume of the feed mixture for 2 h in the feed compartment. We prepared a calibration curve for characterization of the water-isopropanol mixtures by plotting the refractive index measured using an Abbe refractometer (Atago-3T, Tokyo, Japan), against the composition used to prepare each mixture. We performed the PV experiment and determined the percentage of water in the water-isopropanol mixture by measuring the refractive index of the permeate collected in the trap that was suspended inside a liquid nitrogen jar and using the calibration curve. The results reported here are the averages of three PV tests. The results of permeation for water-isopropanol mixtures during the pervaporation were reproducible within an acceptable range.

From the pervaporation data, the separation performance of each of the membranes was estimated in terms of total flux (*J*) using Equation (6):(6)$$J=\frac{W}{A\;t}$$
where *W* is the total weight of permeate (kg) collected in time t (h) and *A* is effective membrane area (m^2^). The separation selectivity, *α_sep_*, was calculated using Equation (7):(7)$$\alpha_{sep}=\frac{P_{w}/P_{IPA}}{F_{w}/F_{IPA}}$$
where *P_w_* and *P_IPA_* are the weight percentages of water and isopropanol in the permeate. *F_w_* and *F_IPA_* are the respective weight percentages of water and isopropanol in the feed.

## 4. Conclusions

Composite membranes were developed by crosslinking sodium alginate with poly(styrene sulphonic acid-co-malic acid) and use of titanium dioxide particles. The membranes prepared were used for the pervaporation separation of water-isopropanol mixtures. A higher level of titanium dioxide content in the membrane resulted in a simultaneous increase of both permeation flux and selectivity. This was explained based on the increase of the size of the amorphous domains and on the establishment of hydrogen bonding in the membrane matrix. The membranes showed significantly lower activation energy values for water permeation than those for isopropanol permeation, indicating that the titanium dioxide nanocomposite membranes have higher separation efficiency towards water. These nanocomposite membranes showed an excellent performance for the removal of water from isopropanol, and this is strictly in accordance with the diffusion data. The evaluation of the heat of sorption, which increased marginally with increasing titanium dioxide content in the membrane, indicated that the hole-filling mechanism is predominant with titanium dioxide due to Langmuir’s mode of sorption.

## Figures and Tables

**Figure 1 molecules-25-01298-f001:**
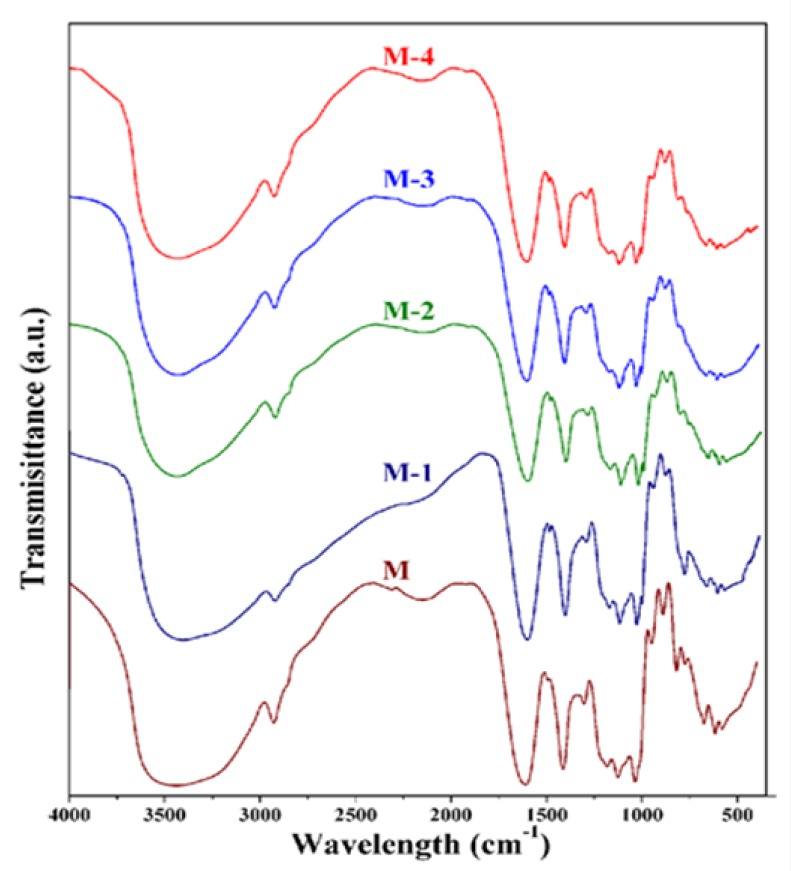
FTIR spectra of crosslinked sodium alginate and related nanocomposite membranes. M: 0%, M-1: 10%, M-2: 20%, M-3: 30%, and M-4: 40% TiO_2_ (all % *w*/*w*).

**Figure 2 molecules-25-01298-f002:**
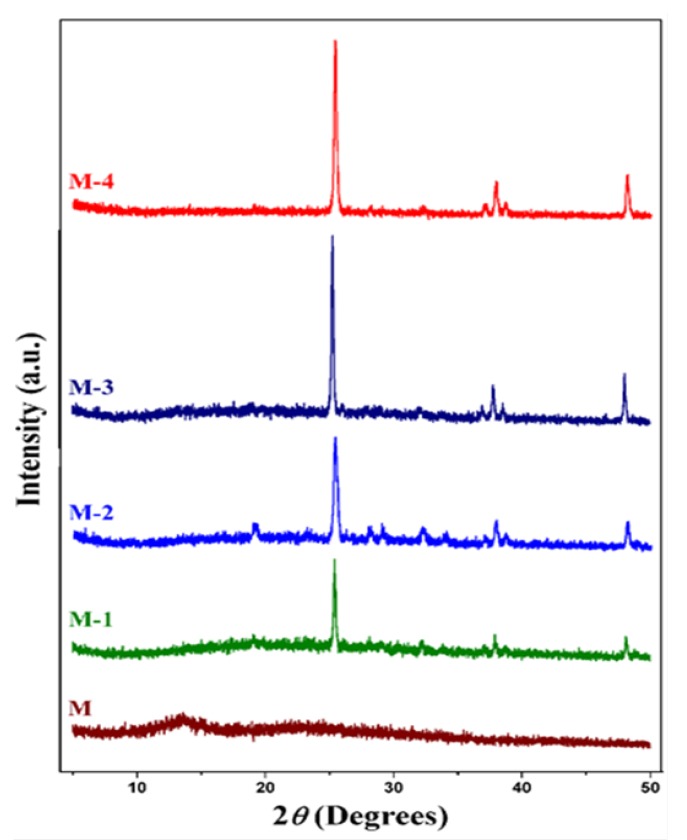
Wide-angle X-ray diffraction patterns of crosslinked sodium alginate and related nanocomposite membranes. M: 0%, M-1: 10%, M-2: 20%, M-3: 30%, and M-4: 40% titanium dioxide (all % *w*/*w*).

**Figure 3 molecules-25-01298-f003:**
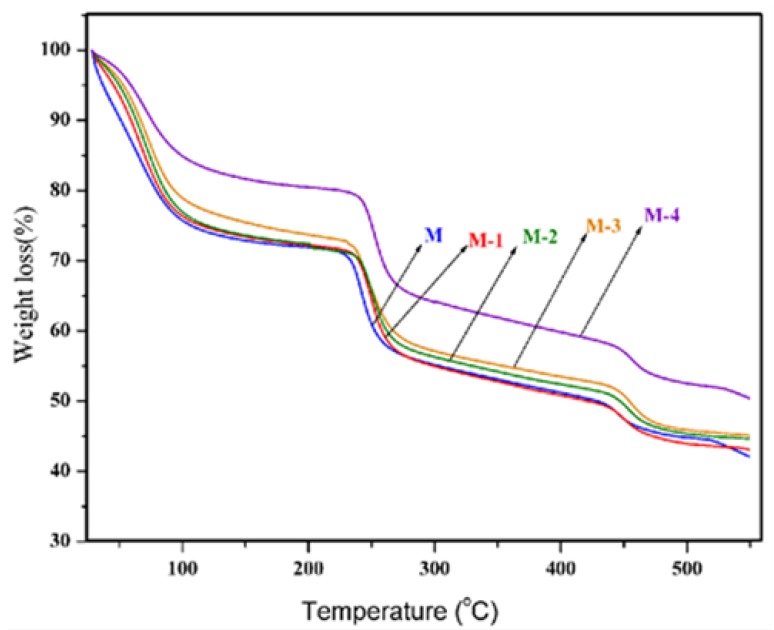
Thermogravimetric analysis of crosslinked sodium alginate and related nanocomposite membranes. M: 0%, M-1: 10%, M-2: 20%, M-3: 30%, and M-4 40% titanium dioxide (all % *w*/*w*).

**Figure 4 molecules-25-01298-f004:**
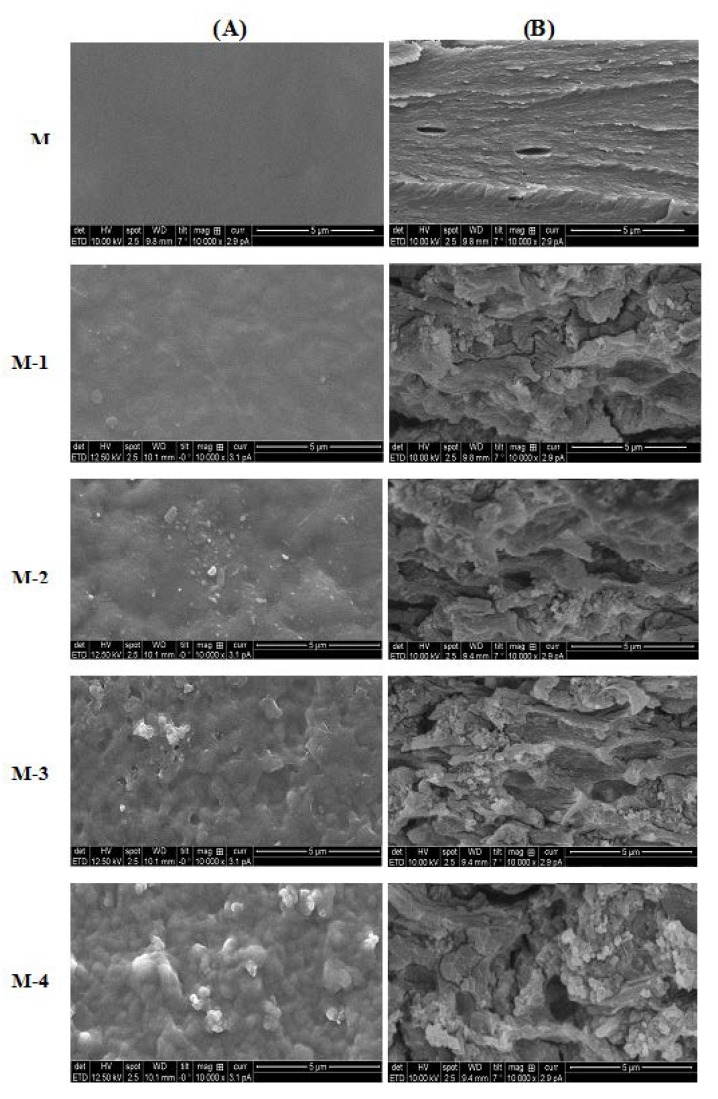
Scanning electron microscopy micrographs of crosslinked sodium alginate and related nanocomposite membranes. (**A**) surface views and (**B**) cross-sectional views.

**Figure 5 molecules-25-01298-f005:**
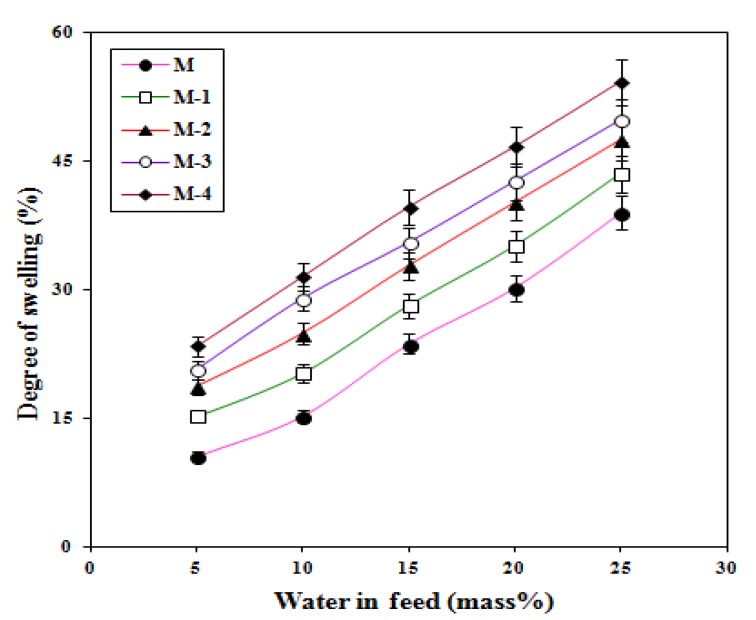
Variation of the degree of swelling with different composition in the feed for crosslinked NaAlg and membranes incorporating TiO_2_.

**Figure 6 molecules-25-01298-f006:**
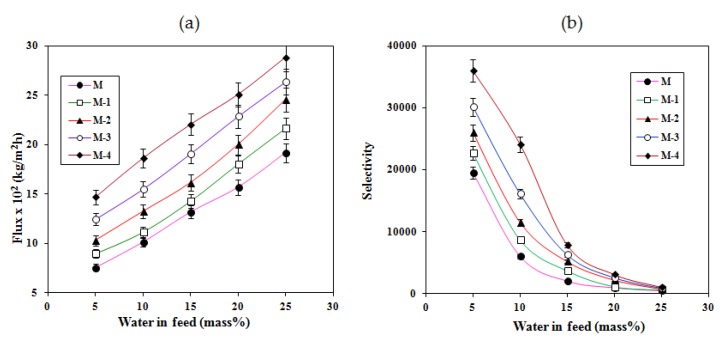
Variation of the (**a**) total pervaporation flux and (**b**) separation selectivity with composition of the feed for crosslinked NaAlg and related membranes containing TiO_2_ at 30 °C.

**Figure 7 molecules-25-01298-f007:**
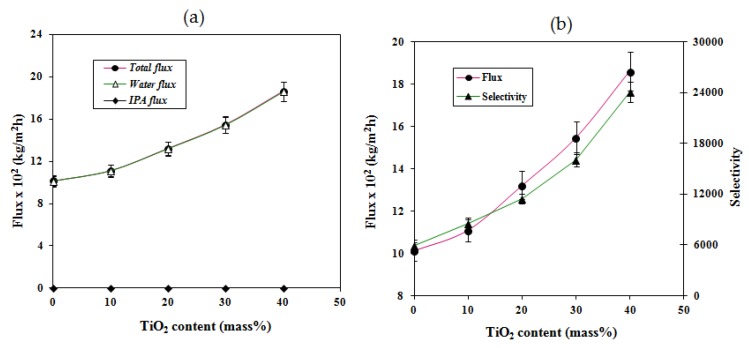
Variation of (**a**) flux and selectivity and (**b**) total flux and the fluxes of water and isopropanol with different percentages of TiO_2_ in the membrane at 10 mass% of water in the feed.

**Figure 8 molecules-25-01298-f008:**
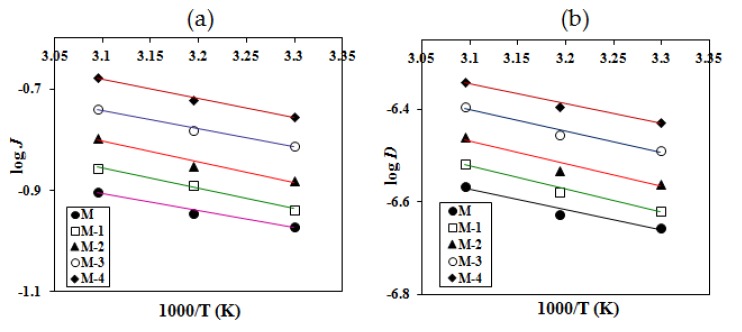
Variation of (**a**) log *J* and (**b**) log *D* with temperature for crosslinked sodium alginate and related nanocomposite membranes with 10% water in the feed.

**Figure 9 molecules-25-01298-f009:**
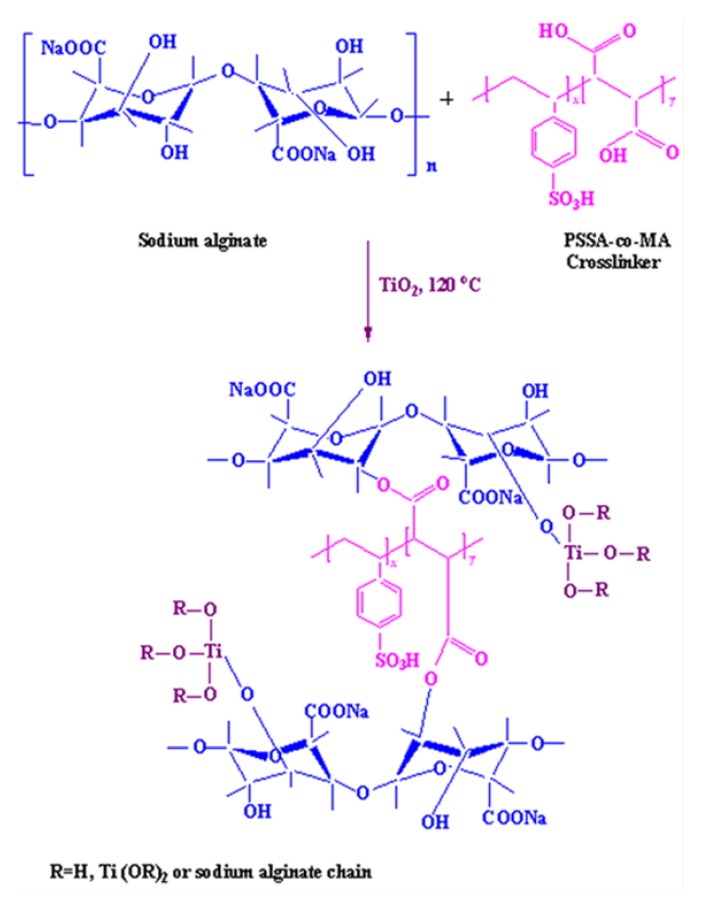
Scheme for the preparation and structural representation of TiO_2_-incorporated crosslinked sodium alginate membranes.

**Figure 10 molecules-25-01298-f010:**
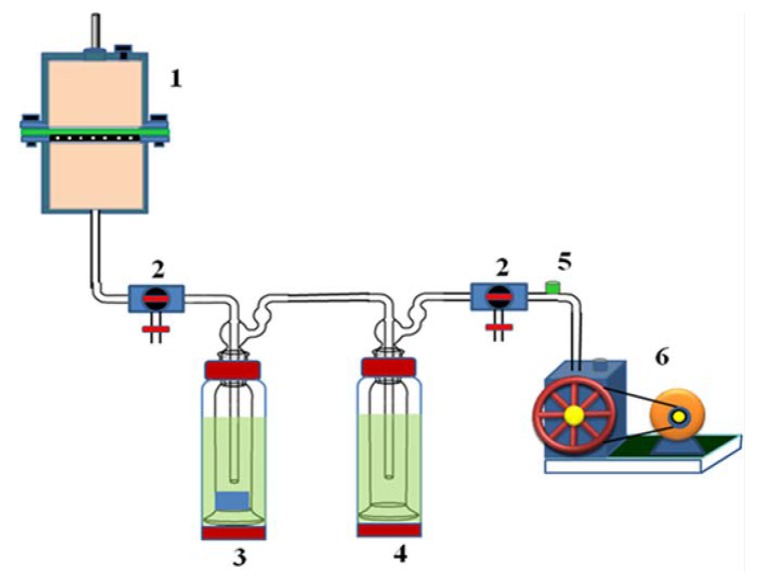
Schematic representation of the pervaporation apparatus: (**1**) pervaporation cell; (**2**) vacuum control valves; (**3**) permeate cold trap; (**4**) moisture cold trap; (**5**) pressure sensor; (**6**) vacuum pump.

**Table 1 molecules-25-01298-t001:** Mechanical properties, Tg, and crystallinity values of crosslinked NaAlg and its TiO_2_ incorporated composite membranes.

Membrane	Tensile Strength (MPa)	Elongation at Break (%)	Glass Transition Temperature (Tg) °C	Crystallinity % *w*/*w*
M	42.83	14.28	240	29.3
M-1	53.21	9.50	244	24.3
M-2	51.50	7.60	246	24.0
M-3	50.43	7.17	248	22.9
M-4	33.67	5.50	250	21.4

**Table 2 molecules-25-01298-t002:** Comparative study of pervaporation performance of sodium alginate-based membranes reported in literature for the dehydration of isopropanol.

Membrane	Thick-ness (µm)	Temp. (^o^C)	Water in the Feed (wt%)	Flux (kg/m^2^∙h)	Separation Factor	PSI	References
NaAlg/PVA (75/25)	35–40	30	10.0	0.06	195	4.87	[37]
Composite membranes of NaAlg and CS	-	30	10.0	0.55	2010	1113	[37]
NaAlg/De (60/40)	35–40	30	10.0	0.07	8991	656	[38]
NaAlg/De (50/50)	35–40	30	10.0	0.10	8172	776	[38]
NaAlg/PAAM-g-GG blend	10	30	10.0	0.13	890	-	[39]
NaAlg + 5 wt% PVA + 10 wt% PEG	10	30	10.0	0.15	3600	-	[40]
MCM-41 (10 wt%) filled NaAlg	10	30	10.0	0.55	30,000	-	[41]
SBA-15 (10 wt%) filled NaAlg	10	30	10.0	0.11	∞	-	[42]
Fe-SBA 15 (10 wt%) filled NaAlg	10	30	10.0	0.18	∞	-	[42]
Na^+^ MMT (10 wt%) filled NaAlg	10	30	10.0	0.25	∞	-	[43]
NaAlg/NaY zeolite (5 wt%)	40	30	10.0	14.19	191	27	[36]
NaAlg/NaY zeolite (15 wt%)	40	30	10.0	16.07	211	32	[36]
NaAlg/NaY zeolite (30 wt%)	40	30	10.0	23.25	272	66	[36]
NaAlg/CS wrapped MWCNTs (0.5 wt%)	50	30	10.0	13.43	929	145	[44]
NaAlg/CS wrapped MWCNTs (1.0 wt%)	50	30	10.0	16.18	1397	226	[44]
NaAlg/CS wrapped MWCNTs (1.5 wt%)	50	30	10.0	19.67	2134	410	[44]
NaAlg/CS wrapped MWCNTs (2.0 wt%)	50	30	10.0	21.76	6420	1400	[44]
SA/ PSSA-co-MA (0.5 wt%)	40	30	10.0	6.38	6800	500	[45]
SA/PSSA-co-MA (1.0 wt%)	40	30	10.0	8.12	8023	750	[45]
SA/PSSA-co-MA (2.0 wt%)	40	30	10.0	9.46	14,894	1300	[45]
NaAlg/PSSA-co-MA (3.0 wt%)	40	30	10.0	10.53	22,312	2415	[45]
Crosslinked NaAlg with 10 wt% TiO_2_	40	30	10.0	11.12	8,591	599	Present work
Crosslinked NaAlg with 20 wt% TiO_2_	40	30	10.0	13.22	11,491	993	Present work
Crosslinked NaAlg with 30 wt% TiO_2_	40	30	10.0	15.47	16,094	1627	Present work
Crosslinked NaAlg with 40 wt% TiO_2_	40	30	10.0	18.61	24,092	2583	Present work

NaAlg: sodium alginate, PVA: poly(vinyl alcohol), CS: chitosan, De: dextrin, PAAM: poly(acrylamide) PEG: poly(ethylene glycol), GG: guar gum, MWCNTs: multiwalled carbon nanotubes, TiO_2_: titanium dioxide, PSSA-co-MA: poly(styrene sulphonic) acid-co-malic acid, Na^+^MMT: sodium montmorillonite, MCM-41 and SBA-15: mesoporous silica, Fe-SBA-15: iron mesoporous silica. The pervaporation separation index (PSI) is the product of the separation index and the flux. The selectivity of a pervaporation membrane for a specific separation is quantified by the dimensionless separation factor α.

**Table 3 molecules-25-01298-t003:** Diffusion coefficients of different membranes for water and isopropanol at different mass% of water in the feed.

Mass % of Water	Diffusion Coefficients of Water (D_w_) ×10^8^ (cm^2^/s)					Diffusion Coefficients of Isopropanol (D_IPA_) ×10^10^ (cm^2^/s)				
	M	M-1	M-2	M-3	M-4	M	M-1	M-2	M-3	M-4
5	3.18	3.56	4.43	4.95	6.53	1.06	0.89	0.72	0.56	0.52
10	2.22	2.31	2.76	3.26	3.74	2.39	2.06	1.96	1.72	1.31
15	1.80	1.96	2.23	2.65	3.09	4.14	3.92	3.80	3.74	3.46
20	1.59	1.72	1.94	2.36	2.61	22.01	21.12	19.01	15.24	8.37
25	1.46	1.58	1.91	2.15	2.38	35.61	34.11	32.80	32.00	23.11

**Table 4 molecules-25-01298-t004:** Pervaporation flux and separation selectivity of different membranes at different temperatures with 10 % water in the feed.

Temperature °C	Flux(J) × 10^2^ (kg/m^2^∙h)					Separation Selectivity(α_sep_)				
	M	M-1	M-2	M-3	M-4	M	M-1	M-2	M-3	M-4
30	10.14	11.12	13.22	15.47	18.61	5992	8591	11,491	16,094	24,092
40	11.63	13.66	14.29	16.13	19.15	3200	4800	4988	6242	7344
50	13.40	14.46	15.12	17.18	20.72	904	1050	2101	2606	2893

**Table 5 molecules-25-01298-t005:** Arrhenius activation parameters for permeation and diffusion, and heat of sorption.

Parameters (kJ/mol)	M	M-1	M-2	M-3	M-4
Total permeation (E_P_)	10.60	6.41	5.55	5.17	3.96
Total diffusion (E_D_)	10.76	6.62	5.80	5.48	4.29
Permeation of water (E_PW_)	9.87	6.33	4.96	4.65	3.49
Diffusion of water (E_DW_)	10.12	7.56	5.27	4.89	3.75
Permeation of isopropanol (E_PIPA_)	93	104	104	110	125
Heat of sorption (∆Hs)	−0.16	−0.21	−0.25	−0.31	−0.33

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
