# Peer review of "Crosslinked Nanocomposite Sodium Alginate-Based Membranes with Titanium Dioxide for the Dehydration of Isopropanol by Pervaporation"

_molecules, 2020, doi:10.3390/molecules25061298_

Round 1
Reviewer 1 Report
the manuscript is well written/cited and can be accepted after some minor revisions:
the introduction should be more critical and highlighting the novelty of the work and why this paper can address some issues. some numerical results must be added in the abstract to give the readers an overall information on the membranes performance. please change SEM images to better quality ones. figs 7-11 should be merged together into 2 figures. the same for figures 12 and 13. the whole manuscript should be checked for language typos.
Author Response
The manuscript is well written/cited and can be accepted after some minor revisions:
We thank the reviewer for their positive comments and suggestions to enhance the ms. We have responded positively to each suggestion
the introduction should be more critical and highlighting the novelty of the work and why this paper can address some issues. some numerical results must be added in the abstract to give the readers an overall information on the membranes performance.
We have modified the manuscript and also included some numerical results in the abstract in line with this suggestion.
please change SEM images to better quality ones.
We spent considerable effort to provide SEM images of high quality, those presented were indeed the best we achieve. We have avoided using artificial image processing to enhance the contrast in what are low contrast samples and we selected the least damaging conditions of the e-beam to avoid any problems with these soft materials. We believe these figures are of sufficient quality to extract the information described in the text.
figs 7-11 should be merged together into 2 figures. the same for figures 12 and 13. the whole manuscript should be checked for language typos.
We have merged these figure as suggested but maintained them as 2 plots as completely merging the figures would make comprehension of the figure rather difficult.
The manuscript has been thoroughly checked by a research for whom English is their first language.
Reviewer 2 Report
Review:
Cross-linked Nanocomposite Sodium Alginate Based Membranes with Titanium Dioxide for the Dehydration of Isopropanol by Pervaporation
This study is fascinating, but some information is missing or not sufficiently explained/discussed. Results are exciting, but, in my opinion, the manuscript needs some revisions before its publication in the journal of Molecules.
In this study, authors prepared sodium alginate-based membranes via a casting solution technique and cross-linked them with poly (styrene sulfonic acid-co-maleic acid) and subsequently modified the membranes by the incorporation of titanium dioxide. The fabricated membranes used to investigate the dehydration of isopropanol by pervaporation.
1- Why 4% w/w aqueous solution of sodium-alginate? Why did you (authors) decide 4% w/w? Is this the optimum percentage based on your previous experiments or other references?
2- What was the thickness of the casted membranes?
3- Figure 6: please provide the scale bar for better understanding the SEM images.
4- Please provide references for the FTIR (page 6, line 193).
5- Please rewrite this part: This indicates that there is a good compatibility between the cross-linked sodium alginate and the titanium dioxide leading to high-quality membranes with an enhancement of the overall pervaporation performance.
6- None of the figures have error bars. Please provide a standard deviation for all the figures.
7- Have you tested the membranes pore size or pore size distribution? If yes, please provide it.
8- One of the reasons for higher selectivity and water permeation was membrane hydrophilicity. But, no contact angle measurement is reported. Then, how the higher hydrophilicity can be approved?
9- Figure 11 shows higher TiO2 resulted in higher flux, in a linear trend, but TiO2 aggregates at a higher concentration that might cause some adverse effects on the flux, have you considered this?
10- Please explain why adding TiO2 resulted in the crystallinity reduction of the membranes?
11- “The membranes showed a significantly lower activation energy values for water permeation…” as you mentioned “significantly” have you done the statistical analysis for this?
12- Please add XRD references, page 7
13- SEM images alone are not enough to show roughness or smoothness of the membranes, have you done any AFM?
14- Is there any explanation for changing the membrane morphology showing in the cross-sectional view of the membranes?
15- Have you tested these membraned for other azeotropic compounds?
Reviewer 3 Report
The paper entitled: Crosslinked Nanocomposite Sodium Alginate Based 2 Membranes with Titanium Dioxide for the 3 Dehydration of Isopropanol by Pervaporation describes interesting research work, but supplementary results and discussion are needed in order to prove the efficiency of the investigated membranes. Thus the authors should address the points outlined below before the final acceptance: 1. English must be improved. Ex. "...experiments was...", "...were prepared using a solution techinique..." - There are many writing errors. Please read carefully the manuscript. ex. lines 72-73 and others. 2. Abstract: - please state in a brief sentence the practical applicability - please include values in comparison with the best reference membrane 3. Please include images of the membranes 4. Please also consider the following references: 1) M only with cross linking agent 2) M only with TiO2 (please include all the existent results (FTIR, SEM, etc...) also for them together with the discussion. In this way, the authors are able to prove the efficiency of M-1 - M-4 and mostly of M-4. 5. The authors should include FTIR, SEM, XRD, DSC also for TiO2. 6. XRD. The peaks at 26 theta degrees, 38, theta degrees are given by the anatase crystalline form of TiO2. Please state this clearly. The phrase: "We attribute ...." lines 225-227 is ambiguous. Please elucidate which forms and bonds!!!! - In the XRD pattern of M appears a protuberance rather than a peak. Peaks are those from M-1 - M-4. The protuberance shows a beginning of crystallization and not a pure crystalline phase. 7. Line 308 - radicals HO· are generated instead of HO- groups. 8. Figs 7 and 8, the authors should prove that the variations are statistically significant. How many replicates have been performed? Please include the variation range. 9. Table 2 - Please explain PSI and separation factor. 10. line 401 - authors should prove that the increase from 2415 to 2583 is statistically significant. 11. Please include a scheme for the mechanism of pervaporation and show the role of the crosslinking agent and of TiO2 in the process. This would be more suggestive.Author Response
The paper entitled: Crosslinked Nanocomposite Sodium Alginate Based Membranes with Titanium Dioxide for the Dehydration of Isopropanol by Pervaporation describes interesting research work, but supplementary results and discussion are needed in order to prove the efficiency of the investigated membranes. We thank the reviewer for identifying the interesting aspects of this work.
We have setout to respond positively to the points raised. In some cases we considered some similar matters whilst we were preparing the manuscript and we pruned the information so only that which is essential to presenting the raw data and its interpretation were included to keep the length of the paper to a manageable length.
- English must be improved. Ex. "...experiments was...", "...were prepared using a solution techinique..." - There are many writing errors. Please read carefully the manuscript. ex. lines 72-73 and others.
The specific lines identified above have been reworded and the complete ms has been reviewed by a native English speaker
- Abstract: - please state in a brief sentence the practical applicability - please include values in comparison with the best reference membrane. We have included numerical values in the abstract. We had already included the application, “the dehydration of isopropanol”.
- Please include images of the membranes We are puzzled by this statement – the ms already includes SEM micrographs so we can only conclude that the reviewer is suggesting light photographs which will convey no additional information and we think could not justify the additional space required.
- Please also consider the following references: 1) M only with cross linking agent 2) M only with TiO2 (please include all the existent results (FTIR, SEM, etc...) also for them together with the discussion. In this way, the authors are able to prove the efficiency of M-1 - M-4 and mostly of M-4.
We have presented the numerical data on the pervaporation performance so we find it difficult to understand what the reviewer means by “the authors are able to prove the efficiency….”
The reviewer proposes the presentation of data on membranes of M only with the cross-linking agent, but M is just that. By M only with TiO2 we sense the reviewer is suggesting M plus TiO2 without the cross-linking agent. This would lead to unstable films which could not be used for pervaporation tests.
We have re read the appropriate sections on the definitions of M and M2, M2, M3 and M4 in the ms to ensure there is no scope for misunderstanding and we could identify no problems.
- The authors should include FTIR, SEM, XRD, DSC also for TiO2. We fail to see the value of including these data as these are tests on a material straight from the supplier.
- XRD. The peaks at 26 theta degrees, 38, theta degrees are given by the anatase crystalline form of TiO2. Please state this clearly. We have done so
The phrase: "We attribute ...." lines 225-227 is ambiguous.
We found the line “Please elucidate which forms and bonds!!!!” not to be understandable. – What is this related too. Difficult to see how any reliable information can be obtained regarding bonds from the limited Q range of data available in the WAXS patterns
“In the XRD pattern of M appears a protuberance rather than a peak. Peaks are those from M-1 - M-4.” The protuberance shows a beginning of crystallization and not a pure crystalline phase. Polymers apart from the unusual systems such as polydiacetylenes prepared by topochemical polymerization fall into the category defined by IUPAC as crystallizable polymers in which there is only part of the polymer involved in the crystal structure giving rise to the so-called semi-crystalline polymers. As a consequence the peaks are broader due to small crystal size, and that displayed in Figure M is very typical of sodium alginate.
- Line 308 - radicals HO· are generated instead of HO- groups. I believe the reviewer is thinking about the photo catalysis of TiO2, which does involve HO radicals but these are very short-lived ~ millisconds and would not lead to the superhydrophilic surface. It may well be that radicals are involved but only as an intermediate step. It is reasonable to identify that a detailed discussion on this mechanism is outside the scope of this ms which is why we provided suitable references. We have rewritten the appropriate lines in the ms.
- Figs 7 and 8, the authors should prove that the variations are statistically significant. How many replicates have been performed? Please include the variation range. These figures now contain “error-bars” and some of the information requested was already included in the text. The modified figures show clearly that the data points shown whether in membrane composition or in the composition of the feedstock are marked and significant.
- Table 2 - Please explain PSI and separation factor. Footnotes added to table
- line 401 - authors should prove that the increase from 2415 to 2583 is statistically significant.
- These values are related to the lower activation energy values obtained for water permeation (EPW) as compared to activation energy values of isopropanol (EPIPA), we concluded that the membranes are highly predominant for water permeation. It is difficult to see how a statistical analysis would add great clarity to the values reported.
- Please include a scheme for the mechanism of pervaporation and show the role of the crosslinking agent and of TiO2 in the process. This would be more suggestive.
The reviewer presents a formidable challenge. We have given considerable though to this challenge. To combine the complex behavior of the membrane and its dynamic behavior in to a single static image would not yield any great insight that is provided by the text of the ms. We believe that we have adequately covered the microscopic role of each component in the text and an image simply reflecting such factors would add nothing to providing an understanding of the process.
Round 2
Reviewer 3 Report
The authors have not considered all the comments outlined in the first review. For a valuable improving of the manuscript, my request is to read carefully each of them and to complete the manuscript. In this way, the article will achieve the high scientific level of the journal.
In this regard, my recommendation is major revision.
Please include images of the membranes
Please also consider the following references:
1. M only with cross linking agent
2. M only with TiO2 (please include all the existent results (FTIR, SEM, etc...) also for them together with the discussion. In this way, the authors are able to prove the efficiency of M-1 - M-4 and mostly of M-4.
3. The authors should include FTIR, SEM, XRD, DSC also for TiO2.
4. Figs 7 and 8, the authors should prove that the variations are statistically significant. How many replicates have been performed? Please include the variation range.
5. Please include a scheme for the mechanism of pervaporation and show the role of the crosslinking agent and of TiO2 in the process. This would be more suggestive.